# Evidence of the Role of Omega-3 Polyunsaturated Fatty Acids in Brain Glucose Metabolism

**DOI:** 10.3390/nu12051382

**Published:** 2020-05-12

**Authors:** Fabien Pifferi, Stephen C. Cunnane, Philippe Guesnet

**Affiliations:** 1Unité Mixte de Recherche (UMR), Centre Nationnal de la Recherche Scientifique (CNRS), Museum National d’Histoire Naturelle (MNHN) 7179, Mécanismes Adaptatifs et Evolution (MECADEV), 1 Avenue du Petit Château, 91800 Brunoy, France; 2Department of Medicine, Université de Sherbrooke, Sherbrooke, QC J1H 5N4, Canada; Stephen.Cunnane@USherbrooke.ca; 3Research Center on Aging, Sherbrooke, QC J1H 4C4, Canada; 4Department of Pharmacology and Physiology, Université de Sherbrooke, Sherbrooke, QC J1H 5N4, Canada; 5PG Consulting, 91440 Bures sur Yvette, France; guesnet07@gmail.com

**Keywords:** n-3 polyunsaturated fatty acids, brain functions, glucose, metabolism, glucose hypometabolism, ageing

## Abstract

In mammals, brain function, particularly neuronal activity, has high energy needs. When glucose is supplemented by alternative oxidative substrates under different physiological conditions, these fuels do not fully replace the functions fulfilled by glucose. Thus, it is of major importance that the brain is almost continuously supplied with glucose from the circulation. Numerous studies describe the decrease in brain glucose metabolism during healthy or pathological ageing, but little is known about the mechanisms that cause such impairment. Although it appears difficult to determine the exact role of brain glucose hypometabolism during healthy ageing or during age-related neurodegenerative diseases such as Alzheimer’s disease, uninterrupted glucose supply to the brain is still of major importance for proper brain function. Interestingly, a body of evidence suggests that dietary n-3 polyunsaturated fatty acids (PUFAs) might play significant roles in brain glucose regulation. Thus, the goal of the present review is to summarize this evidence and address the role of n-3 PUFAs in brain energy metabolism. Taken together, these data suggest that ensuring an adequate dietary supply of n-3 PUFAs could constitute an essential aspect of a promising strategy to promote optimal brain function during both healthy and pathological ageing.

## 1. Introduction

Current knowledge of nutrition and neuroscience demonstrates that nutrients are capable of influencing the development and maintenance of brain function in mammals, including humans. This is well exemplified by nutritional deficiencies occurring during the perinatal period, such as deficiencies in vitamins (B12, folates), essential amino acids (tryptophan), or trace elements (iodine, iron, zinc). Such deficiencies lead to alterations in several cellular processes (myelination, neurotransmission) that alter the normal development of cognitive function in infants [1]. Some fatty acids, in particular polyunsaturated fatty acids (PUFAs) of the n-3 series (also named omega-3 fatty acids), are essential nutrients that help ensure optimal brain and visual development, and help to preserve normal physiology in adulthood [2,3].

Nutrition is not only a factor in optimal brain development but also a major element in maintaining brain function throughout life, particularly during ageing. Studies on nutritional neuroscience suggest that several nutrients are essential to sustain cognitive functions during ageing. These include folic acid [4], antioxidants such as polyphenols [5], and different forms of lipids, such as n-3 PUFAs or molecules derived from them, including ketones [2,6].

In mammals, brain function, and in particular neuronal activity, has high energy needs. Neurotransmission is the most energy-consuming process in the central nervous system. More particularly, it is the restoration of membrane potential by Na^+^/K^+^-exchanging ATPase after an action potential that costs a great deal of energy (consuming 50% of brain ATP, see [7] for review). Glucose is the major oxidative fuel for the brain. When glucose is supplemented by alternative oxidative substrates under different physiological conditions, these fuels do not fully replace the functions fulfilled by glucose [8]. Thus, it is of major importance that the brain is almost continuously supplied with glucose from the circulation. During healthy ageing, brain glucose utilization decreases by approximately 5–10% at approximately 70 years of age, a change that could be accompanied by altered cognitive and behavioral functions. Even if such alterations are often minor in healthy older people, they increase the risk of more serious cognitive decline, as observed during neurodegenerative diseases [9,10,11]. For example, in Alzheimer’s disease (AD), the global decrease in brain glucose utilization easily reaches 25% and is accompanied by severe cognitive impairment [12]. Numerous studies describe the decrease in brain glucose metabolism during healthy or pathological ageing [10,11], but little is known about the mechanisms that cause such impairment. Interestingly, in a study in healthy young subjects, carriers of the allele ε4 of apolipoprotein E (APOE-ε4, a risk factor for AD) exhibited lower brain glucose utilization than non-carriers, years before any other symptom [13]. Cognitive deficits appear much later, suggesting that brain glucose hypometabolism could contribute to declining brain function later in life [12].

Although it appears difficult to determine the exact role of brain glucose hypometabolism during healthy ageing or during age-related neurodegenerative diseases such as AD, uninterrupted glucose supply to the brain is still of major importance for proper brain function. Interestingly, a body of evidence suggests that dietary n-3 PUFAs might play a significant role in brain glucose regulation. Thus, the goal of the present review is to assess this evidence and address the role of n-3 PUFAs in brain energy metabolism.

## 2. Brain Energy Metabolism, Neuronal Activity, Glucose Utilization, and n-3 PUFAs: Early Studies

PUFAs are crucial dietary fatty acids for human health as modulators of the architecture and function of cellular membranes, endogenous mediators of cell signalling and gene expression, and precursors of several enzymatic cascades of lipid mediators [2]. They are derived from two metabolically distinct families, the n-6 and n-3 series. Of these, α-linolenic acid (ALA, 18:3n-3) is the n-3 metabolic precursor of longer-chain eicosapentaenoic acid (EPA, 20:5n-3) and docosahexaenoic acid (DHA, 22:6n-3), which is notably implicated in brain and retina structure and function [2].

Bernsohn and Spitz [14] first suggested the role of α-linolenic acid in brain-membrane-bound glucose-6-phosphatase activity, but the first reports directly addressing the relation between n-3 PUFAs and brain energy metabolism come from studies by Bourre and colleagues [15,16,17]. First, it was demonstrated that the activity of brain Na+/K+-exchanging ATPase, which controls ion homeostasis of synapses, was reduced by 40% in nerve terminals of rats deficient in n-3 PUFAs over several generations, nerve terminals in which the DHA membrane concentration was decreased by 50–80% [15]. In parallel, these rats exhibited significantly lower performance in a learning task. The same research group later demonstrated that long-chain (LC) n-3 PUFA supplementation (fish oil) was also able to modulate Na^+^/K^+^-exchanging ATPase [17]. Three iso-enzymes of Na^+^/K^+^-exchanging ATPase exist: α1, α2, and α3. It has been demonstrated that each of these iso-enzymes has a different affinity for ouabain, a specific inhibitor of Na^+^/K^+^-exchanging ATPase; α1 has low affinity with ouabain, α2 has high affinity, and α3 has very high affinity. In contrast, in rats receiving the fish-oil-supplemented diet (rich in EPA and DHA), the ratio of each iso-enzyme was modified, as was its affinity for ouabain. However, in this study, rats receiving n-3 PUFA supplementation had a only limited effect on brain PUFA content compared to controls, suggesting that the effect of n-3 PUFAs on Na^+^/K^+^-exchanging ATPase was not necessarily due to a direct membrane effect. Numerous subsequent studies then demonstrated the impact of n-3 PUFAs (deficiency or supplementation) on the neurotransmission process itself [18,19], but none of them hypothesized a relationship with energy supply.

A second wave of studies suggested that impaired neurotransmission in animals fed n-3-PUFA-deficient diets could be linked to impaired brain energy metabolism. These studies focused on the impact of n-3 PUFAs on brain glucose utilization and started with rats fed n-3-PUFA-deficient diets over 2 generations. Initially, brain glucose utilization was assessed using the autoradiographic 2-deoxy-glucose (2-DG) method developed by Sokoloff and colleagues [20]. It is known that 2-DG is a non-metabolized analogue of glucose that crosses the blood–brain barrier (BBB) and is transported by specific glucose transporters through the cell membrane. Once in neural cells, it undergoes the first step of the glycolytic pathway: glucose phosphorylation into glucose-6-phosphate by hexokinase. However, further metabolism of 2-DG is then blocked and the deoxy form of glucose-6-phosphate cannot be transported out of the cell by glucose transporters; thus, it accumulates.

With this method, Ximenes and colleagues [21] demonstrated that rats fed an n-3-PUFA-deficient diet exhibited a 50% decrease in glucose utilization in the cerebral cortex and hippocampus compared to rats fed a control diet. Furthermore, the impact of n-3 PUFA deficiency on oxidative phosphorylation was assessed by measuring cytochrome oxidase (CO) activity. CO is the last enzyme of the respiratory chain in the inner mitochondrial membrane and is considered to be a metabolic marker for neuronal functional activity [22]. CO activity was also reduced by 25% to 30% in the same two regions, with a lower glucose utilization rate [21]. Further studies using microarray analysis demonstrated that increasing the n-3 dietary levels (fish oil supplementation in rats) led to a significant increase in the expression of genes coding for energy metabolism enzymes, including CO, NADH dehydrogenase, and ATP synthase [23]. This last study was the first to demonstrate that long-chain n-3 PUFA supplementation could efficiently increase brain energy metabolism by increasing DHA concentrations in neuron membranes.

## 3. n-3 PUFAs and Brain Glucose Transport

Our team then focused on glucose transporter 1 (GLUT1), the main point of glucose entry in the brain. The first generation of adult male rats fed diets lacking n-3 fatty acids was utilized to measure the cerebral cortex glucose transporter protein GLUT1 isoforms of the blood–brain barrier (molecular weight of 55 kDa) and astrocytes (45 kDa) by Western immunoblotting and their mRNA by RT-PCR analysis [24]. The neuronal glucose transporter GLUT3 was also assayed. These analyses were performed using fractions of isolated microvessels and homogenates of the cerebral cortex to identify the different effects on the different isoforms of glucose transporters. The levels of n-6 PUFAs, mainly arachidonic acid (AA, 20:4 n-6), in the phospholipid fractions of microvessels were higher, and the levels of n-3 PUFAs, mainly DHA, were lower than in cerebral cortex homogenates of n-3-fatty acid-deficient rats. The microvessels and cortices of rats fed the n-3-PUFA-deficient diet had DHA levels that were 50% of those in the rats fed the control diet, with DHA replaced by long-chain PUFAs of the n-6 series, 22:5 n-6. The immunoreactivity of the 55-kDa GLUT1 isoform in n-3-PUFA-deficient microvessels was decreased by 25% and the 45 kDa isoform of GLUT1 was decreased by 30% in the homogenates. However, the immunoreactivity of GLUT3 did not change, nor did the amount of GLUT1 mRNA in the rats fed the n-3-PUFA-deficient diet. However, in a more recent study focused on different brain areas, it was reported that n-3 PUFA deficiency specifically repressed GLUT1 gene expression in the frontoparietal cortex in the basal state and also during neuronal activation, which specifically stimulated GLUT1 expression [25]. This suggests that the molecular mechanisms are not yet precisely known.

These results suggest that decreased glucose utilization in the cerebral cortex of n-3-PUFA-deficient rats is linked to reduced amounts of the blood–brain barrier and astrocytic GLUT1 isoforms, and indicates both transcriptional and post-transcriptional regulation of GLUT1 synthesis. Therefore, this study demonstrated that lower brain glucose utilization could be due to a lower transport of glucose across the blood–brain barrier. Thus, we proposed the hypothesis that the impairments of neuronal activity previously observed in n-3-PUFA-deficient animals could result in perturbations of brain energy metabolism, particularly impaired brain glucose utilization. Based on these observations, further work then focused on GLUT expression in the main cell types implicated in brain glucose utilization—endothelial cells, astrocytes, and neurons.

In a complementary study, GLUT1 expression in the cerebral cortex microvessels of rats fed different amounts of n-3 PUFAs (low vs. adequate vs. high) was studied [26]. Western immunoblotting analysis showed that endothelial GLUT1 significantly decreased (−23%) in the n-3-PUFA-deficient microvessels compared to control microvessels, whereas it increased (+35%) in the microvessels of rats fed the high-n-3-PUFA diet. This last result demonstrated that GLUT1 expression was positively correlated with PUFA content in brain microvessels in vivo. In addition, the binding of cytochalasin B (a method used to measure glucose transporter affinity to a specific ligand) indicated that the maximum binding to GLUT1 (Bmax) was reduced in n-3-PUFA-deficient rats. This suggested that n-3 PUFAs modulate brain glucose transport in endothelial cells of the blood–brain barrier, possibly via changes in GLUT1 protein expression and activity. To complement these data on glucose transport to the brain, we tested the hypothesis that glucose transport in brain endothelial cells was correlated with their n-3 PUFA content. We compared the impact of DHA and the other two main LC-PUFAs, AA and EPA, on the fatty acid composition of membrane phospholipids, glucose uptake, and the expression of the 55-kDa GLUT1 isoform in a model of rat brain endothelial cells (RBECs) in primary culture. Without PUFA supplementation, cells became depleted of DHA. After exposure to supplemental AA, EPA, or DHA (15 μM, i.e., a physiological dose), RBECs avidly incorporated these PUFAs into their membrane phospholipids, thereby resembling physiological conditions, i.e., the PUFA content of rat cerebral microvessels. Basal glucose transport in RBECs, measured by a method using [(3)H]-3-o-methylglucose uptake, was increased after EPA or DHA supplementation by 50% and 35%, respectively, whereas it was unchanged with arachidonic acid (AA, 20:4 n-6) supplementation [27]. This increase in glucose transport was associated with increased GLUT1 protein, while GLUT1 mRNA was not affected. Physiological doses of n-3 LC-PUFAs have a direct and positive effect on glucose transport and GLUT1 density in RBECs, which could partly explain the decreased brain glucose utilization in n-3-PUFA-deprived rats [27]. The effects of PUFAs on the expression of GLUT1 at the level of endothelial cells and astrocytes occur through cellular mechanisms that have not yet been elucidated.

Aas and colleagues [28] carried out EPA supplementation experiments on glucose metabolism in skeletal muscle cells. EPA at high concentrations (100–600 μM) stimulated the use of glucose associated with increased GLUT1 mRNA expression without affecting GLUT4. Taken together with the results from the previously mentioned studies [24,25], these results suggest that n-3 PUFAs can modulate basal glucose uptake by specifically altering the expression of GLUT1 transporters in various tissues under differing experimental conditions.

## 4. Positron Emission Tomography Studies

These early studies were then complemented by an in vivo demonstration of the role of n-3 PUFAs in brain glucose metabolism using 18F-fluorodeoxyglucose (FDG)-dynamic positron emission tomography (PET) techniques. Using n-3-PUFA-deficient rats, a significant reduction in 18F-FDG uptake was shown in the whole brain, resulting in both a lower rate of brain uptake during the early phase of the kinetics (0–15 min) and a lower plateau level of brain incorporation during the later plateau phase (15–45 min) [29,30].

Subsequent work was then conducted to address the impact of PUFA supplementation on brain glucose utilization with protocols involving both human and non-human primates, particularly the grey mouse lemur (*Microcebus murinus*), an emerging and promising animal model in the fields of neuroscience and ageing [31,32,33]. In addition to sharing closer phylogenetic proximity to humans than rodents [31,34], the grey mouse lemur also shares several neuroanatomical traits with higher primates, including humans [35,36], and has the advantage of being a true omnivorous species [37], which is of major interest in applied nutrition studies. In 2015, our group described the positive effect of long-chain n-3 PUFAs from fish oil on mouse lemur brain glucose utilization using PET 18F-FDG [38]. In parallel, a behavioral evaluation of the animals was conducted to link both behavioral outcomes to brain glucose metabolism. Lemurs supplemented with n-3 LC-PUFAs exhibited higher brain glucose uptake and cerebral metabolic rate of glucose than controls in all brain regions. The n-3-LC-PUFA-supplemented animals also had higher exploratory activity in an open-field task and less anxiety in the Barnes maze than controls [38,39]. These results demonstrated for the first time in a non-human primate that n-3 LC-PUFA supplementation improved brain glucose uptake and metabolism, while concomitantly reducing anxiety. It is noteworthy that control lemurs were not severely deficient in n-3 PUFAs, as in earlier rodent studies they were fed a control diet with sub-optimal levels of n-3 and n-6 PUFAs. In plasma, fish-oil-supplemented lemurs exhibited a ratio of total n-6 PUFAs/total n-3 PUFAs equal to 0.7:1, whereas the ratio was 4.35:1 in the control group. This ratio should be close to 1:1 according to recommendations for human health [40,41]. In a subsequent study using the same lemurs (further fed with the same diets), we demonstrated that n-3 PUFA supplementation had a significant impact on cognition, electrocortical activity, and neurogenesis [42]. More specifically, lemurs supplemented with fish oil exhibited better learning and memory performance and tended to have lower anxiety levels. The n-3 PUFA supplementation increased the power of alpha, beta, and gamma frequency bands in electrocortical recordings, which are related to various aspects of memory and decision-making [43,44,45,46,47]. Lemurs supplemented with n-3 PUFAs exhibited a higher number of new neurons in brain areas related to memory and emotion than control animals. Altogether, these results point to long-term, significant positive effects of dietary n-3 PUFAs on various functions of the primate brain that are linked to improved brain glucose metabolism. This is particularly the case for electrocortical activity, which is directly linked to neuronal energy supply. Indeed, neuronal activation triggers increased brain glucose consumption and glucose demand, with new glucose being brought in by stimulated blood flow and glucose transport over the blood–brain barrier [48,49].

The impact of n-3 PUFAs on brain glucose utilization was tested after 3 weeks of LC n-3 PUFA supplementation in healthy human volunteers by Cunnane and colleagues [50]. In this study, we used PET 18F-FDG to evaluate whether supplementation with a fish oil rich in n-3 fatty acids increased cerebral glucose metabolism in healthy young or elderly adults. First, the data confirmed that elderly participants had significantly lower cerebral glucose entry into the brain than the young participants, supporting the interest in developing strategies to support brain energy supply during ageing. However, there was no effect of n-3 PUFA supplementation on glucose metabolism in any of the brain regions studied, a result that could be attributed to the very short period of n-3 PUFA dietary supplementation.

## 5. Potential Mechanisms of Action

Some potential mechanisms can explain the impact of PUFAs on brain glucose metabolism. One can envisage a direct effect of membrane incorporation of PUFAs on the ability of GLUT proteins to transport glucose. Indeed, it has been proposed that the intrinsic properties of the transmembrane protein GLUT1 can be modified to adjust glucose uptake. This possibility has been described mostly in peripheral tissues [49] but was also observed in astrocytes. The incubation of astrocytes with glutamate provokes a fast increase in glucose uptake through a the maximal uptake (Vmax) rate and is an intrinsic component, together with affinity (Km), of the transporter characteristics. Since n-3 PUFA dietary manipulation significantly modifies the composition of the lipid bilayer in which transmembrane proteins such as GLUT are inserted, we can hypothesize that it consequently affects the intrinsic properties of these proteins.

Other examples exist of a similar relationship. For example, EPA or DHA incorporation in the membrane of aortic endothelial cells induces an increase in plasma membrane fluidity [51]. Mitchell and colleagues demonstrated that the activity of G-protein-coupled signalling depends on the level of unsaturation in membranes (with a higher level of membrane unsaturation leading to optimized protein activity) [52].

PUFAs can also modulate brain glucose utilization through less direct pathways. In particular, an effect on local cerebral blood flow, which is directly correlated with local cerebral activity [49], can also be proposed because it is modulated by the intake of n-3 PUFAs. Indeed, the passage of glucose through the BBB is directly correlated with the local utilization of glucose, which is itself linked to local cerebral blood flow [20,49]. This flow can be increased by n-3 PUFA dietary supplementation. In aged monkeys (18 years old) fed for 1 to 4 weeks on a diet supplemented with DHA, an increase in local cerebral blood flow (+27%) was measured by PET imaging [53]. Moreover, dietary 22:6 n-3 supply restored the age-related impairment of the coupling between neuronal activation and regional cerebral blood flow (CBF) response to vibrotactile stimulation in the monkey brain cortex [53]. We also know that the dietary intake of long-chain n-3 PUFAs in the form of fish oils rich in EPA and DHA modifies hemodynamic parameters, such as blood pressure [54]. In addition, chronic dietary deficiency in n-3 PUFAs increases blood pressure in adult rats, which is probably linked to a defect in the regulation of blood volume via angiotensin II [55]. Among the mechanisms involved downstream, the decrease in the amount of GLUT1 transporter in animals deficient in n-3 PUFAs may also be the consequence of a decrease in energy production (ATP) by brain tissue. Indeed, the mitochondrial activity of cytochrome C oxidase is reduced in animals deficient in n-3 PUFAs, highlighting a possible deficit in ATP production [21]. Similarly, the expression of mRNAs of many enzymes of the mitochondrial oxidative phosphorylation pathway is affected by the variation in dietary intake of n-3 PUFAs [23], while the activity of Na^+^/K^+^-exchanging ATPase is greatly reduced in n-3-PUFA-deficient animals [17].

Interestingly, the effect of n-3 PUFA dietary deficiency appears to be specific to glucose transport and metabolism, particularly in brain endothelial cells. Indeed, this deficiency does not alter the transport of amino acids at the BBB level (L-phenylalanine and alpha-aminoisobutyrate) [56]. In contrast, in this study, permeability to sucrose was increased, suggesting changes in the permeability properties of the barrier involving tight junctions, the expression of which could be modulated by n-3 PUFAs (through the modification of occludin mRNA expression) [57].

However, the expression of the neural transporter GLUT3 does not seem to be impacted by n-3 PUFA deficiency. GLUT3 has a much higher affinity for glucose than GLUT1 (by a factor of 7), and the cerebral extracellular glucose concentration (approximately 2 mM) is systematically higher than the Km of GLUT3 (<1 mM), suggesting that GLUT3 expression is not a limiting step during variations in cerebral glucose consumption (during neuronal activation or hypoglycemia, for example) [58,59]. Only chronic hypoglycemia (the infusion of insulin for 1 week) increases the density of neuronal GLUT3, which then decreases the cerebral local use of glucose. This mechanism would allow neurons to maintain their energy metabolism in a situation of extracellular glucose deficit [60].

At the cellular level, several mechanisms can be proposed to explain the decrease in the amount of GLUT1 in n-3-PUFA-deficient animals. Recent data suggest a possible effect of n-3 PUFA deficiency on energy metabolism through an impact of peroxisome proliferator-activated receptors (PPARs) [61]. Indeed, n-3 PUFA deficiency modified the mRNA levels of PPARs and mitochondrial uncoupling proteins (UCPs) in the juvenile rat brain. More precisely, n-3 PUFA deficiency affected the mRNA levels of PPARrα, PPARβ/δ, and PPARγ in the cerebral cortex and the cerebellum, and UCP5 mRNA levels were 30% lower in the cerebellum of n-3-PUFA-deficient rats, confirming that the impact of n-3 PUFAs on energy metabolism could be found at the molecular and mitochondrial levels. In the specific case of glucose transporters, a possible modification of the expression of GLUT mRNAs by PUFAs (through their role as natural agonists of PPARs) is ruled out, since no change was observed in the level of expression of GLUT1 mRNAs by real-time quantitative PCR [24]. This suggests that the regulation of GLUT1 takes place at the post-transcriptional or perhaps post-translational level. The post-transcriptional regulation of GLUT1 in the brain has already been observed through the modification of the quantity of mRNA during hypoglycemia or hypoxia [62,63], or by modifying the stability of mRNAs (the modification of the 3′-region not transcribed by specific proteins) [58,64,65]. A post-translational regulation has also been identified involving the glycosylation of the transporter GLUT1 and its translocation at the level of the plasma membrane [66,67]. In addition, numerous studies suggest that changes in the quantity of GLUT1 in the endothelial cells of the BBB (especially during hypoglycemia) are accompanied by a change in the distribution of the transporter between the different compartments of the cell (luminal membrane, abluminal membrane, cytoplasm), thereby regulating glucose transport to the BBB [68,69]. Therefore, our results obtained in vivo demonstrate that the dietary intake of n-3 PUFAs (deficit) decreases the absolute quantity of GLUT1 in brain endothelial cells and astrocytes [24,26].

## 6. Conclusions

Glucose is the fundamental fuel supporting neurotransmission and cerebral functions [8], and it is now widely accepted that PUFAs constitute essential nutritional factors to allow proper brain development and functioning throughout life [70]. The studies included in the present review support the hypothesis that n-3 PUFAs are essential factors for optimal brain glucose metabolism, since they participate in regulating key steps in the delivery of blood glucose to neural cells. Their effects have been observed at the organ level (whole-brain glucose entry and metabolism measured by PET imaging), at the cellular level (on glucose transporters in endothelial cells, and astrocytes and Na^+^/K^+^-exchanging ATPase in neurons), at the mitochondrial level (with effects on UCP, NADH dehydrogenase, cytochrome oxidase, ATP synthase, etc.), and at the molecular level (PPARs). At all these levels, the impacts of PUFAs come from both the modification of the lipid composition of cell membranes supporting the proteins involved in all these mechanisms and from precursors of molecular messengers.

Since normal and neuropathological ageing is associated with brain glucose hypometabolism and increased risk of cognitive impairment [9,11], we suggest here that ensuring an adequate dietary supply of n-3 PUFAs is an essential aspect of strategies to promote optimal brain function during ageing. Several studies in humans (reviewed in [71]) show that high dietary intake of long-chain n-3 PUFAs could contribute significantly to the reduction of the risk of AD and cognitive decline. Therefore, it is tempting to propose the hypothesis that dietary intake of n-3 PUFAs might exert protective effects against the risk of AD or other forms of dementia by modulating brain energy and glucose metabolism.

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
