# Peer review of "Evidence of the Role of Omega-3 Polyunsaturated Fatty Acids in Brain Glucose Metabolism"

_nutrients, 2020, doi:10.3390/nu12051382_

Round 1

Reviewer 1 Report

This review addresses changes in brain tissue glucose uptake and/or metabolism that results from modified n-3 PUFA dietary challenges to animals or n-3 PUFA challenges to brain endothelial cells in 2D tissue culture. The authors are fair and balanced in their review of the field and limit self-citations to ~12 of their key contributions (out of 72 references). Overall the review adds a nice synopsis of the impact of n-3 PUFA on brain energy balance to the field. The succinct mechanistic speculation in the closing section is nicely stated. Overall a nice, useful review.

There are a few minor points to raise:

  1. It is 12 pages of text without figures or summary cartoons of mechanisms. Was the lack of all figures deliberate, even a summary pathway of potential mechanisms or a table of potential PUFA impact sites of glucose metabolism?
  2. Is there room here for a short discussion of energy from lipids vs glucose? It is key to many organs under stress and may be valid here in the context of disease pathophysiology - it could be brief - not required, just a point of balance if brain energy is the center of this discussion.
  3. Do you wish to include a brief paragraph on n-3 uptake in brains in animals or humans vs the in vitro results? Passage of n-3 through the BBB requires special use of lysophospholipids for transport, which could impact some interpretation of comparative studies, where such limitations on 2D cell culture do not apply. The PET studies are nice discussion that led me to these thoughts.
  4. Please clarify the speculation on hypothetical transcriptional control in line 286 having just stated in line 285 that most regulation was suspected to be post-transcriptional and post-translational. Just seemed like a bit of a change in direction for that series of thoughts.
  5. Minor typographical errors or noun/verb mismatches occur in
    1. lines 61 (need to insert "to" or "with")
    2. Line 61 mismatched plurals for nouns and verbs
    3. Line 88 "However in"
    4. line 96 has an author internal notation as text?
    5. line 172 insert "in" or "with"
    6. line 196 check spacing
    7. Line 224 check text
    8. Line 274 spacing needs adjustment

Author Response

We would like to thank the reviewers and the editor for their encouraging and fruitful comments. We took them into account to produce a new version of the manuscript. In addition, the last version of the MS has been revised by an editing company, with the certificate available as attachment.

This review addresses changes in brain tissue glucose uptake and/or metabolism that results from modified n-3 PUFA dietary challenges to animals or n-3 PUFA challenges to brain endothelial cells in 2D tissue culture. The authors are fair and balanced in their review of the field and limit self-citations to ~12 of their key contributions (out of 72 references). Overall the review adds a nice synopsis of the impact of n-3 PUFA on brain energy balance to the field. The succinct mechanistic speculation in the closing section is nicely stated. Overall a nice, useful review.

There are a few minor points to raise:

  1. It is 12 pages of text without figures or summary cartoons of mechanisms. Was the lack of all figures deliberate, even a summary pathway of potential mechanisms or a table of potential PUFA impact sites of glucose metabolism?

The absence of figure was deliberate. The review reports various quantity of potential impact/roles of PUFA on brain glucose utilization, at several levels (organs, tissues, cells, and even at the nuclear/molecular level), what is extremely difficult to summarize in a figure. We also had the feeling that a table will not bring additional information and will be insufficient to summarize such mechanisms.

  1. Is there room here for a short discussion of energy from lipids vs glucose? It is key to many organs under stress and may be valid here in the context of disease pathophysiology - it could be brief - not required, just a point of balance if brain energy is the center of this discussion.

Since the goal of the present review was to focus on glucose, and since our group already published reviews on the energy from lipids vs glucose to fuel the brain (see references 1 and 2 below for example), we have the feeling that it will not add pertinent information to the present manuscript.

  1. Do you wish to include a brief paragraph on n-3 uptake in brains in animals or humans vs the in vitro results? Passage of n-3 through the BBB requires special use of lysophospholipids for transport, which could impact some interpretation of comparative studies, where such limitations on 2D cell culture do not apply. The PET studies are nice discussion that led me to these thoughts.

Thank you for your encouraging comment about the PET studies. Concerning the uptake of n-3 in brain, we have the feeling that this question is out of topic in the present review.

  1. Please clarify the speculation on hypothetical transcriptional control in line 286 having just stated in line 285 that most regulation was suspected to be post-transcriptional and post-translational. Just seemed like a bit of a change in direction for that series of thoughts.

That part of the discussion was not very clear in the previous version of the MS, and, according to the reviewers suggestion, we modified it with the following statement (Lines 283-289):

This suggests that the regulation of GLUT1 takes place at the post-transcriptional or perhaps post-translational level. The post-transcriptional regulation of GLUT1 in the brain has already been observed through the modification of the quantity of mRNA during hypoglycaemia or hypoxia63,64 or by modifying the stability of mRNAs (the modification of the 3'-region not transcribed by specific proteins)59,65,66. A post-translational regulation has also been identified involving the glycosylation of the transporter GLUT1 and its translocation at the level of the plasma membrane67,68.

  1. Minor typographical errors or noun/verb mismatches occur in
    1. lines 61 (need to insert "to" or "with")
    2. Line 61 mismatched plurals for nouns and verbs
    3. Line 88 "However in"
    4. line 96 has an author internal notation as text?
    5. line 172 insert "in" or "with"
    6. line 196 check spacing
    7. Line 224 check text : we could not find the problem, please be more specific – English editing might have solved this error?
    8. Line 274 spacing needs adjustment

Apart point 7, all these errors have been corrected, thanks a lot.

Reviewer 2 Report

This is a well written systematic review on a topic of importance in the nutritional neuroscience field focussing on the effects of essential omega-3 (n-3) poly-unsaturated fatty acid (PUFA) to counteract deficits in glucose metabolism, instrumental in fostering brain health. The review is well structured and informative. However, I would recommend the authors to include a paragraph that describes dysregulated glucose metabolism following brain injury and dietary omega-3 fatty acids as a strategy to help the brain to cope with the metabolic depression. This further piece of information could broaden the scope of this manuscript and even inspire exciting future work in this area.

Minor points:

Page 2, Line 88. The words 'However in' is misspelled as ‘Howevein’.

Page 7, Line 306. (NADH-dehydrogenase, cytochrome oxidase, ATP synthase…). The sentence is incomplete.

Author Response

This is a well written systematic review on a topic of importance in the nutritional neuroscience field focussing on the effects of essential omega-3 (n-3) poly-unsaturated fatty acid (PUFA) to counteract deficits in glucose metabolism, instrumental in fostering brain health. The review is well structured and informative. However, I would recommend the authors to include a paragraph that describes dysregulated glucose metabolism following brain injury and dietary omega-3 fatty acids as a strategy to help the brain to cope with the metabolic depression. This further piece of information could broaden the scope of this manuscript and even inspire exciting future work in this area.

Minor points:

Page 2, Line 88. The words 'However in' is misspelled as ‘Howevein’. Done

Page 7, Line 306. (NADH-dehydrogenase, cytochrome oxidase, ATP synthase…). The sentence is incomplete.

This sentence was complete, it is an enumeration of the levels of action of PUFAs. Since the last point was including “…at a molecular level”, we changed it to “…at the molecular level” which fits the rest of the enumeration and finishes the sentence more correctly.

  1. Can ketones compensate for deteriorating brain glucose uptake during aging? Implications for the risk and treatment of Alzheimer's disease. Cunnane SC, Courchesne-Loyer A, St-Pierre V, Vandenberghe C, Pierotti T, Fortier M, Croteau E, Castellano CA. Ann N Y Acad Sci. 2016 Mar;1367(1):12-20. doi: 10.1111/nyas.12999. Epub 2016 Jan 14. Review.
  2. Ketones and brain function: possible link to polyunsaturated fatty acids and availability of a new brain PET tracer, 11C-acetoacetate. Pifferi F, Tremblay S, Plourde M, Tremblay-Mercier J, Bentourkia M, Cunnane SC. Epilepsia. 2008 Nov;49 Suppl 8:76-9. doi: 10.1111/j.1528-1167.2008.01842.x.